# An Improved YOLOv5s-Seg Detection and Segmentation Model for the Accurate Identification of Forest Fires Based on UAV Infrared Image

Kunlong Niu [1,2], Chongyang Wang [1], Jianhui Xu [1], Chuanxun Yang [1], Xia Zhou [1,*] and Xiankun Yang [2]

[1] Key Lab of Guangdong for Utilization of Remote Sensing and Geographical Information System, Guangdong Open Laboratory of Geospatial Information Technology and Application, Guangdong Engineering Technology Center for Remote Sensing Big Data Application, Guangzhou Institute of Geography, Guangdong Academy of Sciences, Guangzhou 510070, China; 2112101068@e.gzhu.edu.cn (K.N.); wangchongyang@gdas.ac.cn (C.W.); xujianhui306@gdas.ac.cn (J.X.); yangchuanxun19@mails.ucas.ac.cn (C.Y.)

[2] School of Geography and Remote Sensing, Guangzhou University, Guangzhou 510006, China; yangxk@gzhu.edu.cn

\* Correspondence: zhouxia@gdas.ac.cn

**Abstract:** With the influence of climate change and human activities, the frequency and scale of forest fires have been increasing continuously, posing a significant threat to the environment and human safety. Therefore, rapid and accurate forest fire detection has become essential for effective control of forest fires. This study proposes a Forest Fire Detection and Segmentation Model (FFDSM) based on unmanned aerial vehicle (UAV) infrared images to address the problems of forest fire occlusion and the poor adaptability of traditional forest fire detection methods. The FFDSM integrates the YOLO (You Only Look Once) v5s-seg, Efficient Channel Attention (ECA), and Spatial Pyramid Pooling Fast Cross-Stage Partial Channel (SPPFCSPC) to improve the detection accuracy of forest fires of different sizes. The FFDSM enhances the detection and extraction capabilities of forest fire features, enabling the accurate segmentation of forest fires of different sizes and shapes. Furthermore, we conducted ablation and controlled experiments on different attention mechanisms, spatial pyramid pooling (SPP) modules, and fire sizes to verify the effectiveness of the added modules and the adaptability of the FFDSM model. The results of the ablation experiment show that, compared to the original YOLOv5s-seg model, the models fused with the ECA and SPPFCSPC achieve an improved accuracy, with FFDSM showing the greatest improvement. FFDSM achieves a 2.1% increase in precision, a 2.7% increase in recall, a 2.3% increase in mAP@0.5, and a 4.2% increase in mAP@0.5:0.95. The results of the controlled experiments on different attention mechanisms and SPP modules demonstrate that the ECA+SPPFCSPC model (FFDSM) performs the best, with a precision, recall, mAP@0.5, and mAP@0.5:0.95 reaching 0.959, 0.870, 0.907, and 0.711, respectively. The results of the controlled experiment on different fire sizes show that FFDSM outperforms YOLOv5s-seg for all three fire sizes, and it performs the best for small fires, with a precision, recall, mAP@0.5, and mAP@0.5:0.95 reaching 0.989, 0.938, 0.964, and 0.769, respectively, indicating its good adaptability for early forest fire detection. The results indicate that the forest fire detection model based on UAV infrared images (FFDSM) proposed in this study exhibits a high detection accuracy. It is proficient in identifying obscured fires in optical images and demonstrates good adaptability in various fire scenarios. The model effectively enables real-time detection and provides early warning of forest fires, providing valuable support for forest fire prevention and scientific decision making.

**Keywords:** forest fire detection; infrared image; YOLOv5s-seg; fire segmentation



## 1. Introduction

As an integral part of the Earth's ecosystem, forests possess immeasurable natural and social value and are recognized as the "lungs of the Earth". However, due to the impacts of

global climate change and human activities, forest fires are becoming increasingly frequent, causing significant losses [1–3] and becoming one of the greatest threats to forest security.

Traditional forest fire monitoring is typically conducted through manual patrols, lookout towers, and satellite technologies. However, manual patrols and lookout towers require significant manpower and resources [4]. In recent decades, multi-source satellite remote sensing data with varying spatial resolutions have shown great potential in the detection and monitoring of forest fires, including MODIS (Moderate-resolution Imaging Spectroradiometer), VIIRS (Visible infrared Imaging Radiometer), Landsat 8, Sentinel-2A/B, and so on. Pourshakouri et al. [5] proposed an improved contextual algorithm for detecting small and low-intensity fires based on the MODIS Level 1B Radiance Product, which outperforms the contextual traditional algorithms and has potential for global applications. Comparing the 1-km MODIS and 375-m VIIRS fire products, Fu et al. [6] found that VIIRS outperformed MODIS in detecting forest fires, and showed higher detection accuracy. Ding et al. [7] proposed a deep learning algorithm for wildfire detection based on Himawari-8 data, which greatly improves the detection accuracy compared to traditional machine learning algorithms. However, these higher than 100-m remote sensing data may be more suitable for large-scale fire detection, and showed poor performance for accurately detecting small fires. With 10-m Sentinel-2A\B and 30-m Landsat 8 imageries, Achour et al. [8] mapped the summer forest fires in Tunisia in 2017. The results showed that Sentinel-2 performed better than Landsat 8 in characterizing the forest fires because of its higher spatial resolution. Furthermore, Waigl et al. [9] evaluated three fire detection methods using EO-1 (Earth Observing-1) Hyperion data, which can detect up to 5 m$^2$ high-temperature fires. However, this remote sensing imagery may be difficult to detect small fires because of coarse resolution and cloud cover [10]. In addition, since forest fires often occur in remote mountainous areas, the aforementioned methods face significant challenges in accurately monitoring and responding to fires on time [11].

In recent years, with the rapid development of unmanned aerial vehicle (UAV) technology, monitoring and response efforts for forest fires have greatly improved [4,12,13]. By leveraging the UAV's maneuverability, high resolution, and fast information transmission capabilities, the comprehensive, real-time, and efficient monitoring of forest fires can be achieved.

Traditional methods for forest fire detection rely on image-processing techniques, which primarily detect the occurrence of fires based on features such as smoke and flame color, shape, and texture. Celik et al. [14] proposed a color model for smoke detection based on the YCbCr (Luminance, Chrominance-Blue, Chrominance-Red) color space, but this method had a high false detection rate. Chen et al. [15] analyzed the dynamic patterns of fire growth and disorder using the RGB (Red, Green, Blue) model, ultimately achieving fire detection based on the RGB color space, which involved simpler calculations than other color spaces. Toreyin et al. [16] used spatial wavelet transform of the current image and background image to monitor the reduction in high-frequency energy in the scene, thus detecting smoke. However, this method's accuracy may be affected by cloud cover. Borges et al. [17] extracted features such as flame color, surface roughness, centroid height ratio, and flame flicker frequency and used a Bayesian classifier for discrimination, achieving good classification results. Gubbi et al. [18] proposed smoke detection based on a wavelet transform and support vector machine classifier, which yielded satisfactory results but had a slower processing speed.

Throughout its development, deep learning has been applied to forest fire detection in three main ways: image classification, object detection, and image segmentation. Among these techniques, the methods based on object detection are the most commonly used because they offer higher accuracy and ease of use compared to image classification and segmentation. Object detection methods can be categorized into two-stage detection models, represented by the R-CNN (Region-based Convolutional Neural Networks) series [19,20], and one-stage detection models, represented by the YOLO (You Only Look Once) series [21] and SSD (Single Shot Multibox Detector) [22]. Compared to two-stage



detection models, one-stage detection models achieve faster detection speeds while maintaining excellent accuracy, and among the one-stage detection models, the YOLO series outperforms SSD [23–25]. Srinivas et al. [26] proposed the application of a basic CNN (Convolutional Neural Networks) architecture for classifying forest fire images, achieving a classification accuracy of 95%. Kinaneva et al. [27] and Barmpoutis et al. [24] used the Faster R-CNN algorithm to detect smoke and flames in UAV images. Jiao et al. [28,29] proposed modified versions of YOLOv3-tiny and YOLOv3 for the real-time detection of flames and smoke in drone images.

However, the aforementioned forest fire detection methods are based on optical images. Although detection methods based on such images are widely used, they have limitations in fog and low-light conditions and are prone to obstruction. Therefore, in this study, UAV infrared images are adopted to detect the obstructed forest fire. A Forest Fire Detection and Segmentation Model (FFDSM) is proposed based on the YOLOv5s-seg model. The FFDSM model incorporates the Efficient Channel Attention (ECA) [30] module and the Spatial Pyramid Pooling Fast Cross-Stage Partial Channel (SPPFCSPC) [31] module, enhancing the accuracy of fire detection and the capability to extract forest fire features. This model addresses the issue of the poor adaptability of traditional methods.

## 2. Materials and Methods

### 2.1. Dataset and Processing

The FLAME (Fire Luminosity Airborne-based Machine learning Evaluation) dataset [32] is a forest fire dataset comprising aerial images captured via UAV, produced by Northern Arizona University in the United States. The dataset includes optical videos recorded using UAV cameras as well as infrared videos recorded using infrared thermal imagers (https://ieee-dataport.org/open-access/flame-dataset-aerial-imagery-pile-burn-detection-using-drones-uavs, accessed on 16 January 2023). The details of the utilized UAVs and cameras are shown in Table 1.

**Table 1.** Description of hardware and tools in the FLAME dataset.

| Hardware and Tools | Details |
|---|---|
| DJI Phantom 3 Pro | wight: 1280 g; diagonal size: 350 mm; max speed: 57.6 km/h; max flight time: 23 min |
| DJI Matrice 200 | wight: 3.8 kg; diagonal size: 643 mm; max speed: 61.2 km/h; max flight time: 27 min |
| DJ Zenmuse X4S | sensor: 1″ CMOS (Complementary Metal-Oxide Semiconductor); focal length: 8.8 mm; FOV (Field of View): 84°; resolution: 1280 × 720; spectral bands: 680–800 nm |
| DJ Phantom 3 | sensor: 1/2.3″ CMOS; focal length: 20 mm; FOV: 94°; resolution: 3840 × 2160; spectral bands: 680–800 nm |
| FLIR Vue Pro R | sensor: Uncooled VOx Microbolometer; focal length: 6.8 mm; FOV: 45°; resolution: 640 × 512; spectral bands: 7.5–13.5 μm |

The FLAME dataset was collected by the fire managers from the Flagstaff (Arizona) Fire Department. They collected four optical videos using Zenuse X4S and Phantom3 cameras and created the videos into 254 × 254 frames for classification and segmentation. In addition, they collected three different palettes of infrared videos using FLIR cameras. The specific information is shown in Table 2.

The FLAME dataset only provides fire classification and segmentation labels for optical images. In this study, we processed the infrared videos in the FLAME dataset to obtain the corresponding images. The steps included the following:

(1) Converting the videos into image format to ensure high-quality image data and conducting quality screening to remove poor-quality images.

(2) Applying data augmentation techniques to the selected images to expand the dataset, including rotation, saturation enhancement, contrast enhancement, and horizontal and vertical flips (results shown in Figure 1). In the end, 5250 infrared images were obtained. The dataset was split into training, validation, and testing sets with a ratio of 8:1:1.

(3) Creating labels for the processed and filtered images to generate txt files for YOLOv5s-seg labeling.

**Table 2.** Specific information of the FLAME dataset.

| Number | Type | Camera | Palette | Duration | Resolution | FPS | Application | Usage | Labeled |
|---|---|---|---|---|---|---|---|---|---|
| 1 | Video | Zenmuse | Normal | 966 s | 1280 × 720 | 29 | Classification | — | N |
| 2 | Video | Zenmuse | Normal | 399 s | 1280 × 720 | 29 | — | — | N |
| 3 | Video | FLIR | WhiteHot | 89 s | 640 × 512 | 30 | — | — | N |
| 4 | Video | FLIR | GreenHot | 305 s | 640 × 512 | 30 | — | — | N |
| 5 | Video | FLIR | Fusion | 25 min | 640 × 512 | 30 | — | — | N |
| 6 | Video | Phantom | Normal | 17 min | 3840 × 2160 | 30 | — | — | N |
| 7 | Frame | Zenmuse | Normal | 39,375 frames | 254 × 254 | — | Classification | Train/Val | Y |
| 8 | Frame | Phantom | Normal | 8617 frames | 254 × 254 | — | Classification | Test | Y |
| 9 | Frame | Phantom | Normal | 2003 frames | 3480 × 2160 | — | Segmentation | Train/Val/Test | Y(Fire) |
| 10 | Mask | — | Binary | 2003 frames | 3480 × 2160 | — | Segmentation | Train/Val/Test | Y(Fire) |

Note: —represents not applicable. N and Y represent NO and YES, respectively.

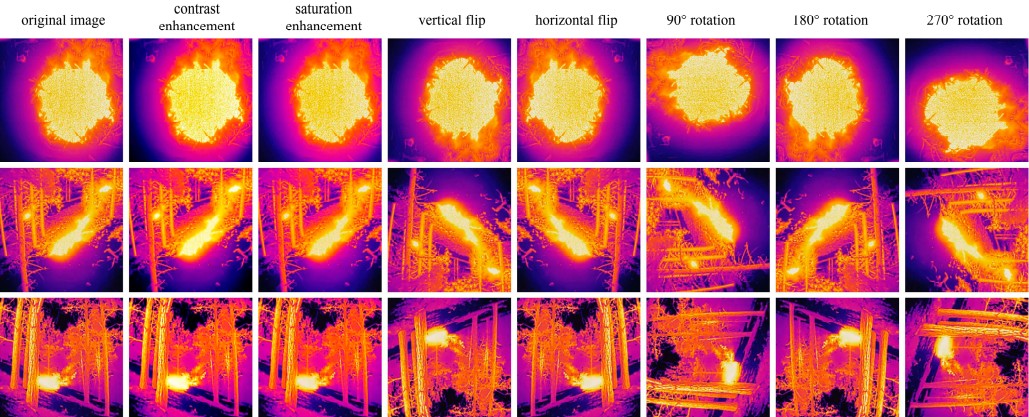

**Figure 1.** Original images and image augmentation results.

*2.2. FFDSM (Forest Fire Detection and Segmentation Model)*

To improve the accuracy of forest fire detection, combined with UAV infrared images, we proposed FFDSM based on the YOLOv5-seg model, as shown in Figure 2. The FFDSM model integrates the ECA module and SPPFCSPC module into the YOLOv5-seg model to further enhance the segmentation accuracy of forest fires. The FFDSM mainly consists of three parts: YOLOv5-seg, the ECA module, and the SPPFCSPC module. By incorporating attention mechanism modules such as ECA, we can effectively capture crucial features in the images and adaptively adjust the weights of the feature maps to enhance the expression and discriminative ability of the targets. On the other hand, spatial pyramid pooling (SPP) modules such as SPPFCSPC enable feature extraction on different scales, allowing for more comprehensive perception and precise object localization. These modules contribute to improving the overall performance and accuracy of the forest fire detection model.

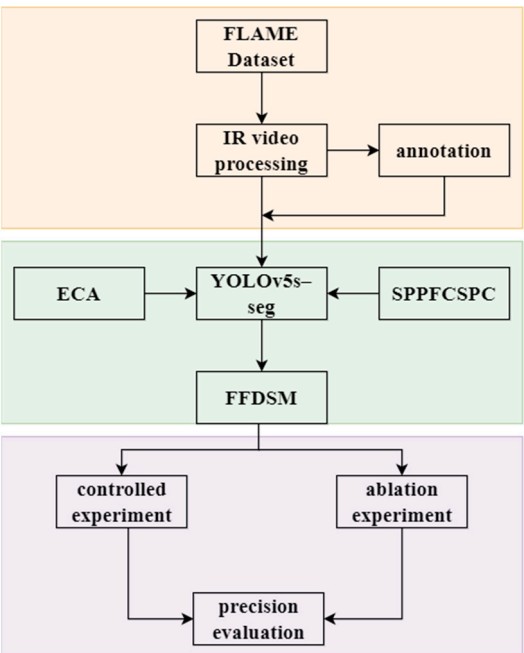

**Figure 2.** The technical process of the FFDSM. (IR video: Infrared video; ECA: Efficient Channel Attention; SPPFCSPC: Spatial Pyramid Pooling Fast Cross-Stage Partial Channel.).

2.2.1. Network Architecture of FFDSM

The network architecture of the FFDSM (as shown in Figure 3) is based on YOLOv5-v7.0 [33], a mainstream, general-purpose object detection method. YOLOv5 comes in four variants, S (small), M (medium), L (large), and X (extra-large), each offering a balance between performance and speed. It has significant advantages when deployed in small devices.

The FFDSM comprises four parts: Input, Backbone, Neck, and Head. An ECA module is added after the ninth layer, and the SPPF (Spatial Pyramid Pooling Fast) module is replaced with the SPPFCSPC module.

The Input mainly includes functionalities such as Mosaic data augmentation, adaptive anchor box calculation, and adaptive image scaling. This stage typically involves image preprocessing, which involves resizing the input images to the network's input size and performing normalization operations. During the network training phase, YOLOv5 utilizes Mosaic data augmentation to enhance the training speed and improve the network's accuracy. It also introduces an adaptive anchor box calculation and an adaptive image-scaling method.

The Backbone mainly uses modules such as CBS, CSP [34] and SPPF to extract image features and continuously shrink the feature map. The Backbone is divided into three modules: the CBS (Cov, BN, SiLU), CSP (Cross Stage Partial) [34], and SPPF. In YOLOv5, the CBS module encapsulates Cov (Convolutional), BN (Batch Normalization), and SiLU (Activation Function). Cov performs dimensionality reduction on feature maps; BN normalizes each batch of data; SiLU is an activation function that increases the nonlinearity of the data. In the YOLOv5-v7.0 version, Leaky ReLU (Activation Function) is replaced with SiLU as the activation function. Compared to the ReLU function, the SiLU function has a smoother curve near zero. Due to its use of the sigmoid function, SiLU performs better than Leaky ReLU in some applications. The CSP1_X structure is applied in the Backbone, while the CSP2_X structure is applied in the Neck. CSP is an important module for feature extraction. Furthermore, in YOLOv5-v7.0, the SPP module is replaced with the SPPF module. The SPPF module utilizes multiple small-sized pooling kernels in a cascade instead of a single large-sized pooling kernel in the SPP module. This preserves the original functionality while further improving the running speed. By incorporating different receptive fields of feature maps and enhancing their expressive power, the model achieves an improved speed.

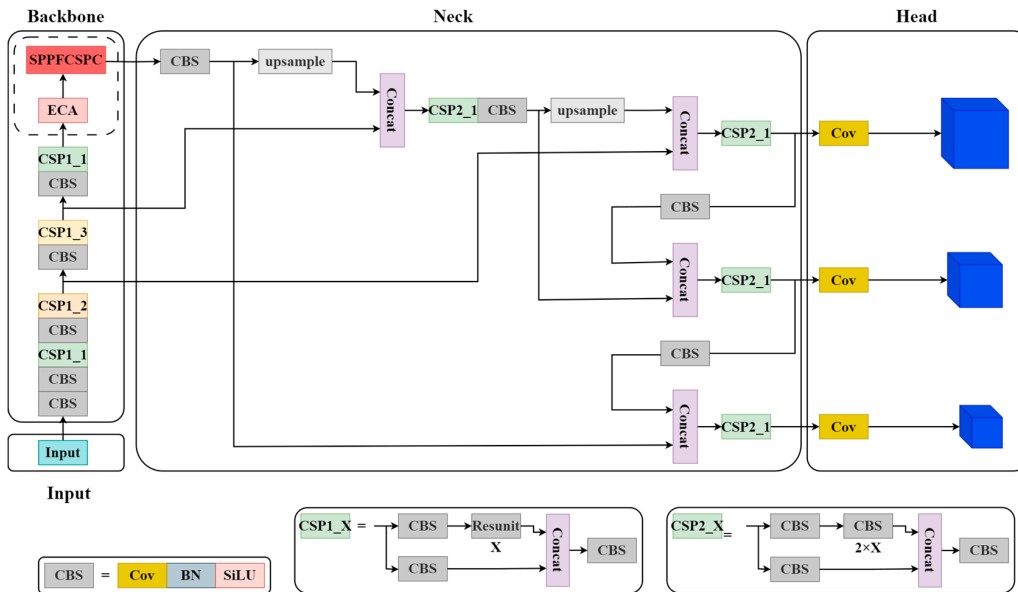

**Figure 3.** The network architecture of the FFDSM. The Input utilizes mosaic data augmentation, adaptive anchor box calculation, and adaptive image scaling for image preprocessing; The Backbone mainly uses modules such as CBS and CSP to extract image features and continuously shrink the feature map. The CBS module consists of a convolutional layer (Cov), a batch normalization layer (BN), and an activation function layer (SiLU). The CSP module consists of three CBS modules and several residual blocks (Resunit); the Neck obtains relatively shallow features from Backbone and then fuses them with deeper semantic features; the Head is composed of three 1 × 1 convolutional layers, detecting three objects of different sizes.

The Neck obtains relatively shallow features from the Backbone and then fuses them with deeper semantic features. The current version of YOLOv5 utilizes the FPN (Feature Pyramid Network) [35] + PAN (Pyramid Attention Network) [36] structure in its Neck. This combination leverages the FPN layer to propagate strong semantic features in a top-down manner, while the feature pyramid conveys strong localization features in a bottom-up manner, which aggregates parameters from different backbone layers to different detection layers.

The Head predicts objects through loss function. The Head primarily improves the GIOU (Generalized Intersection over Union) loss function [37] during training and CIOU (Complete Intersection over Union) function [38] for prediction box filtering. These improvements enhance the accuracy and precision of the model in object detection tasks.

### 2.2.2. ECA (Efficient Channel Attention)

ECA is an attention mechanism module used for image processing [30]. It primarily regulates attention on image channels to enhance the effectiveness of image feature representation. The structure of ECA is shown in Figure 4.

The ECA enhances the attention of convolutional neural networks to different channels in input feature maps through global average pooling and channel weighting. Global average pooling performs its operation on each channel, merging the characteristic values of each channel into a single numerical value. Next, a linear transformation is applied to the global average of each channel to obtain the weights for the channels. These weights are used to adjust the importance of each channel. Compared to other attention models, the advantages of the ECA mechanism lie in its low model complexity, high computational efficiency, and effective results. Therefore, it is widely applied in various fields, such as image classification, object detection, and image segmentation.

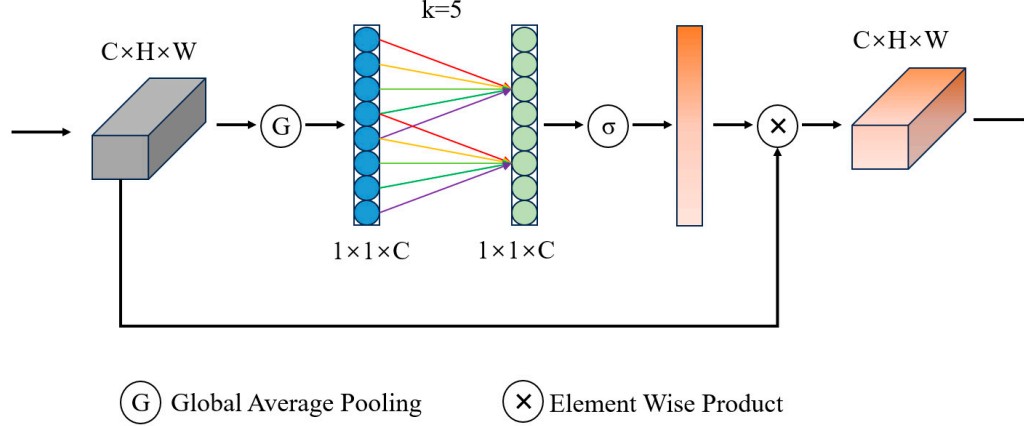

**Figure 4.** The structure of the ECA module. The ECA module extracts global information from input feature maps through Global Average Pooling and Linear Transformation. Subsequently, channel weighting is performed (adaptive selection of kernel size) and the weighted feature map is multiplied by the original feature map to obtain the final feature map. ($C \times H \times W$: image size; k: adaptive selection of kernel size; σ: sigmoid activation function; $1 \times 1 \times C$: feature vectors.).

### 2.2.3. SPPFCSPC (Spatial Pyramid Pooling Fast Cross-Stage Partial Channel)

The SPPFCSPC is an improved version of the SPPCSPC (Spatial Pyramid Pooling Cross-Stage Partial Channel). The SPPCSPC module is the SPP module used in YOLOv7 [31]. It incorporates multiple parallel MaxPool (Maximum Pooling) operations in a sequence of convolutions to avoid the image distortion issues caused by image-processing operations. It also addresses the problem of extracting repetitive features in convolutional neural networks. In the MaxPool module, the MaxPool operation expands the receptive field of the current feature map. Then, MaxPool integrates the results with the feature information processed through standard convolution, thereby enhancing the network's generalization. The structure of the SPPCSPC is illustrated in Figure 5.

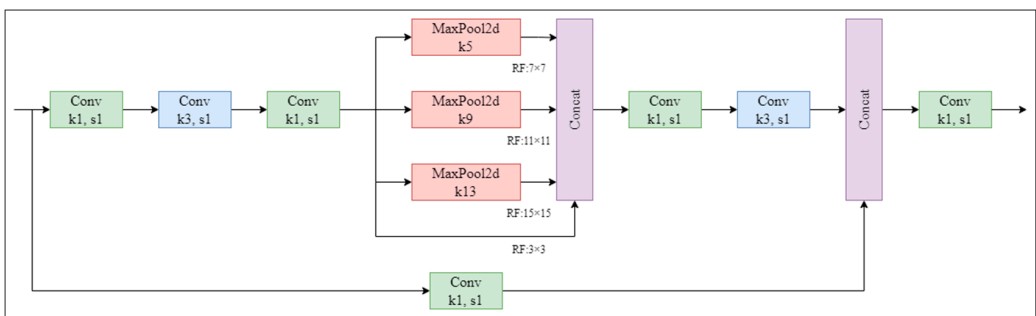

**Figure 5.** The structure of the SPPCSPC module. SPPCSPC performs a series of convolutions on the feature map, followed by maximum pooling and fusion of different receptive fields ($3 \times 3$, $7 \times 7$, $11 \times 11$, $15 \times 15$). After further convolution, it is fused with the original feature map. (Conv: Convolutional; MaxPool2d: Maximum Pooling).

The SPPFCSPC incorporates the concept of improving and upgrading the SPP module to the SPPF module in YOLOv5 and makes further improvements to the SPPCSPC module, resulting in improved speed while keeping the receptive field unchanged. The three different maximum poolings in SPPCSPC have been changed to the same maximum pooling. Next, the three maximum poolings are sequentially connected. The structure of the SPPFCSPC is illustrated in Figure 6.

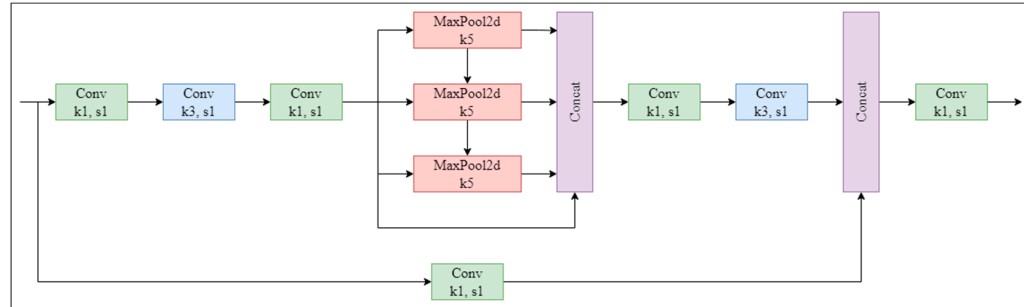

**Figure 6.** The structure of the SPPFCSPC module. SPPFCSPC performs a series of convolutions on the feature map, followed by maximum pooling and fusion of four receptive fields (one $3 \times 3$ and three $7 \times 7$). After further convolution, it is fused with the original feature map. (Conv: Convolutional; MaxPool2d: Maximum Pooling).

*2.3. Experimental Design*

In this study, we evaluated different model components and investigated their impacts on performance through ablation and controlled experiments.

2.3.1. Ablation Experiments

We evaluated the impacts of different modules on performance by gradually removing them from the model. This included individually removing the ECA and SPPFCSPC. The ablation experiments consisted of four models (shown in Table 3).

**Table 3.** Ablation experiments.

| Model | YOLOv5s-Seg | ECA | SPPFCSPC |
|---|---|---|---|
| YOLOv5s-seg | √ | — | — |
| YOLOv5s-seg+ECA | √ | √ | — |
| YOLOv5s-seg+SPPFCSPC | √ | — | √ |
| FFDSM | √ | √ | √ |

2.3.2. Controlled Experiments

In order to gain a deeper understanding of the effects of different attention mechanisms, pyramid modules, and fire sizes on model performance, we conducted experiments using different attention mechanisms, pyramid modules, and fire sizes (considering a fire size smaller than 1/9 of the image size as a small fire, and the others as large fires). The specific experimental design is outlined in Table 4.

**Table 4.** Design of the controlled experiments.

| Controlled Experiments | Exp 1 | Exp 2 | Exp 3 | Exp 4 | Exp 5 | Exp 6 |
|---|---|---|---|---|---|---|
| Different attention mechanisms | ECA | SE [39] | CBAM [40] | CA [41] | — | — |
| Different pyramid modules | SPPFCSPC | SPPCSPC | SPPF | — | — | — |
| Different fire sizes | YOLOv5s-seg+fire-s | FFDSM+fire-s | YOLOv5s-seg+fire-sl | FFDSM+fire-sl | YOLOv5s-seg+fire-l | FFDSM+fire-l |

Note: The terms "fire-s", "fire-sl", and "fire-l" are used to represent different sizes of fires. "Fire-s" stands for a small fire, "fire-sl" represents a combined fire with small and large fires, and "fire-l" signifies a large fire; — represents not applicable; ECA, SE, CBAM, CA represents Efficient Channel Attention, Squeeze and Excitation, Convolutional Block Attention Module, and Coordinate Attention, respectively.

### 2.3.3. Training Environment

The training environment is described in Table 5. The main parameter settings for model training are presented in Table 6.

**Table 5.** Model training environment.

| Training Environment | Details |
|---|---|
| Programming language | Python 3.9 |
| Operating system | Windows 10 |
| Deep learning framework | Pytorch 1.13 |
| GPU | NVIDIA GeForce GTX 1080Ti |

**Table 6.** Parameter settings for model training.

| Training Parameters | Details |
|---|---|
| Epochs | 300 |
| Batch size | 16 |
| img size | $640 \times 640$ |
| Initial learning rate | 0.01 |
| Optimization algorithm | SGD (Stochastic Gradient Descent) |
| Pre-training weight file | None |

### 2.4. Model Evaluation

In this study, we evaluated the performance of the model using precision (P), recall (R), mAP@0.5, and mAP@0.5:0.95. These metrics were used to assess the accuracy, completeness, and overall performance of the model [42–44].

$$P = \frac{TP}{TP + FP} \tag{1}$$

$$R = \frac{TP}{TP + FN} \tag{2}$$

*TP* represents samples where the true class is positive, and the model predicts it as positive. *FP* represents samples where the true class is negative, but the model incorrectly predicts it as positive. *FN* represents samples where the true class is positive, but the model incorrectly predicts it as negative.

$$AP = \int_0^1 P(r)dr \tag{3}$$

In Equation (3), $P(r)$ represents the P–R curve, where the horizontal axis is recall, and the vertical axis is precision. *AP* is the average precision, which represents the area under the P–R curve. It is calculated as the area enclosed by the curve and the axes in the P–R graph:

$$mAP = \frac{1}{N} \sum_{i=1}^{N} AP_i \tag{4}$$

where mAP@0.5 refers to the average precision when the IOU (Intersection over Union) threshold is set to 0.5. On the other hand, mAP@0.5:0.95 represents the average precision when the IOU threshold varies gradually from 0.5 to 0.95. A higher mAP@0.5 indicates a higher detection accuracy of the object detection model for the given dataset. When the value of mAP@0.5:0.95 is high, the algorithm produces accurate detection results at different thresholds, covering a wide range of scenarios and accommodating various application needs.

## 3. Results

### 3.1. Ablation Experiments

In order to validate the impacts of the different improvement modules on the performance of the forest fire detection model, the trained model was tested on the test set to obtain the corresponding evaluation metrics, and the results were analyzed. The results are shown in Table 7.

**Table 7.** Results of the ablation experiments.

| Model | P | R | mAP@0.5 | mAP@0.5:0.95 |
|---|---|---|---|---|
| YOLOv5s-seg | 0.938 | 0.843 | 0.884 | 0.669 |
| YOLOv5s-seg+ECA | 0.949 | 0.860 | 0.898 | 0.694 |
| YOLOv5s-seg+SPPFCSPC | 0.951 | 0.862 | 0.895 | 0.694 |
| FFDSM | **0.959** | **0.870** | **0.907** | **0.711** |

Note: Bold numbers represent perform best in each metrics.

According to the results in Table 7, the models with the ECA module and SPPFCSPC module outperform the YOLOv5s-seg baseline. Among them, the FFDSM model performs the best, with a P of 0.959, R of 0.870, mAP@0.5 of 0.907, and mAP@0.5:0.95 of 0.711. Compared to YOLOv5s-seg, FFDSM shows improvements of 2.1% in P, 2.7% in R, 2.3% in mAP@0.5, and 4.2% in mAP@0.5:0.95. FFDSM achieves the most significant improvement in mAP@0.5:0.95.

Additionally, the model with the SPPFCSPC module performs better in P and R compared to the model with the ECA module. However, these models show opposite performances in terms of mAP@0.5, and both models perform similarly in terms of mAP@0.5:0.95.

### 3.2. Comparison of Different Attention Mechanisms

By introducing attention mechanisms, one can better focus on crucial information, thereby improving detection efficiency and accuracy. In order to select the optimal attention mechanism to enhance the detection performance, we considered several commonly used attention mechanism modules, including CA, CBAM, SE, and ECA. The results are shown in Table 8.

**Table 8.** Comparison results of different attention mechanisms.

| Model | P | R | mAP@0.5 | mAP@0.5:0.95 |
|---|---|---|---|---|
| ECA+SPPFCSPC | **0.959** | 0.870 | **0.907** | **0.711** |
| SE+SPPFCSPC | **0.959** | 0.869 | 0.906 | 0.708 |
| CBAM+SPPFCSPC | 0.951 | 0.866 | 0.899 | 0.701 |
| CA+SPPFCSPC | 0.956 | **0.871** | 0.905 | 0.704 |

Note: ECA, SE, CBAM, CA represents Efficient Channel Attention, Squeeze and Excitation, Convolutional Block Attention Module, and Coordinate Attention, respectively. Bold numbers represent perform best in each metrics.

According to Table 8, among the four attention modules, CBAM+SPPFCSPC performs the worst. On the other hand, ECA+SPPFCSPC performs the best, achieving a P of 0.959, R of 0.870, mAP@0.5 of 0.907, and mAP@0.5:0.95 of 0.711. The next best-performing module is SE+SPPFCSPC, which has the same P as ECA+SPPFCSPC but a slightly lower R, mAP@0.5, and mAP@0.5:0.95. Additionally, CA+SPPFCSPC has the highest recall, reaching 0.871.

### 3.3. Comparison of Different Spatial Pyramid Pooling Modules

In this study, we replaced the SPPF module with the SPPFCSPC module, which is an improvement on the SPPCSPC module. Therefore, we compared the SPPF, SPPFCSPC, and SPPCSPC. The results are shown in Table 9.

**Table 9.** Comparison results of different spatial pyramid pooling modules.

| Model | P | R | mAP@0.5 | mAP@0.5:0.95 |
|---|---|---|---|---|
| ECA+SPPFCSPC | **0.959** | **0.870** | **0.907** | **0.711** |
| ECA+SPPCSPC | 0.957 | 0.864 | 0.901 | 0.698 |
| ECA+SPPF | 0.949 | 0.860 | 0.898 | 0.694 |

Note: Bold numbers represent perform best in each metrics.

Table 9 shows that ECA+SPPFCSPC performs the best across all the evaluation metrics, with a P, R, mAP@0.5, and mAP@0.5:0.95 reaching 0.959, 0.870, 0.907, and 0.711, respectively. Although ECA+SPPCSPC does not perform as well as ECA+SPPFCSPC, it still shows some improvement compared to ECA+SPPF, while ECA+SPPF performs the worst.

*3.4. Comparison of Different Forest Fire Sizes*

In order to assess the suitability of the FFDSM for different fire sizes in the images, we selected images from the test set containing large fires (210 images), small fires (180 images), and a mixture of both large and small fires (150 images). The results are shown in Table 10.

**Table 10.** Comparison results for different fire sizes.

| Size | P | | R | | mAP@0.5 | | mAP@0.5:0.95 | |
|---|---|---|---|---|---|---|---|---|
| | YOLOv5s-Seg | FFDSM | YOLOv5s-Seg | FFDSM | YOLOv5s-Seg | FFDSM | YOLOv5s-Seg | FFDSM |
| small | 0.983 | **0.989** | 0.925 | **0.938** | 0.953 | **0.964** | 0.716 | **0.769** |
| small + large | 0.940 | 0.961 | 0.737 | 0.754 | 0.779 | 0.796 | 0.551 | 0.586 |
| large | 0.940 | 0.951 | 0.816 | 0.842 | 0.854 | 0.874 | 0.597 | 0.644 |

Note: Bold numbers represent perform best in each metrics.

For all three fire sizes, the FFDSM outperforms YOLOv5s-seg. Specifically, in the case of small fires, both YOLOv5s-seg and FFDSM perform the best across all the evaluation metrics. The FFDSM achieves a P of 0.989, R of 0.938, mAP@0.5 of 0.964, and mAP@0.5:0.95 of 0.769. In the case of a mixture of large and small fires, YOLOv5s-seg and the FFDSM demonstrate a better P than they do for large fires. However, in terms of R, mAP@0.5, and mAP@0.5:0.95, the performance is inferior to that in the scenario with large fires.

Furthermore, the FFDSM achieves significant improvements compared to YOLOv5s-seg. In the case of a mixture of large and small fires, the most significant improvement is observed in P, with an increase of 2.1%. In the case of large fires, the most significant improvements are seen in R and mAP@0.5, with increases of 2.6% and 2.0%, respectively. In the scenario with small fires, the most significant improvement is observed in mAP@0.5:0.95, with an increase of 5.3%.

Overall, the FFDSM shows significant improvements over YOLOv5s-seg across different evaluation metrics for different fire sizes. Specifically, the FFDSM performs the best in the case of small fires, achieving excellent results in terms of P, R, mAP@0.5, and mAP@0.5:0.95.

**4. Discussion**

Forest fires are prone to obstruction by trees. In the case of latent fires, smoldering conditions may be caused by forest litter [45]; these fires do not involve open flames or dense smoke and are difficult to detect through optical images. In contrast, infrared cameras measure the thermal radiation emitted by objects, and the captured infrared images can effectively complement optical images. In the optical image (Figure 7a), the forest fire within the red box is obscured by trees or smoke, making it difficult to detect its exact location. In the infrared image (Figure 7b), although the forest fire within the target box is partially obstructed, it can still be captured and detected (Figure 7c).

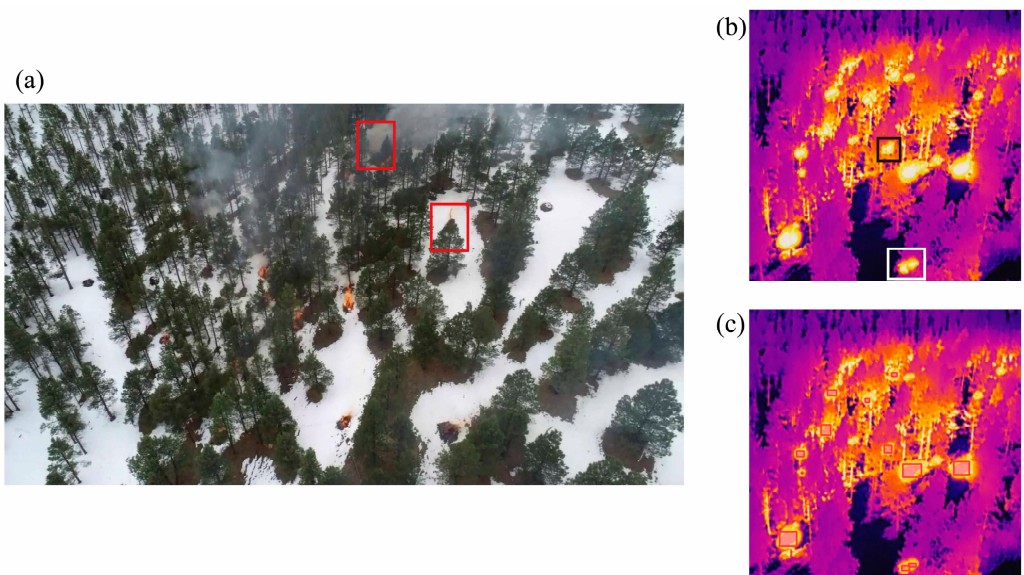

**Figure 7.** Optical image and infrared image: (**a**) optical image; (**b**) infrared image. (**c**) Detection results of the FFDSM for the infrared image. Red boxes in (**a**), white and black boxes in (**b**): obscured fires; Red boxes in (**c**): detection results for obscured fires.

While some researchers have used infrared images for forest fire detection in recent years, most of them have employed traditional image processing methods [46] or used infrared images as an aid in conjunction with optical images for forest fire discrimination [47–49], making it challenging to achieve accurate real-time detection. Based on deep learning methods, the above problems can be effectively solved through direct learning from infrared images in different scenarios. Among the different deep-learning-based forest fire detection methods, object-detection-based methods are the most commonly used due to their high accuracy and usability compared to object classification and segmentation methods [50]. Among the single-stage detection models, YOLOv5 stands out, as it has a small model size, high detection accuracy, and fast detection speed, making it suitable for real-time forest fire detection [10,43,51,52].

In order to further verify the superiority of the FFDSM model, we compared the FFDSM model with several improved forest fire detection methods based on YOLOv5. According to Table 11, Model1 performed the best in terms of P, reaching a value of 0.961. However, it showed the worst performance in terms of R, mAP@0.5, and mAP@0.5:0.95. On the other hand, Model2 had the lowest P (0.947), but it outperformed Model1 in terms of R, mAP@0.5, and mAP@0.5:0.95. Model3 demonstrated a better performance in terms of R, mAP@0.5, and mAP@0.5:0.95 compared to both Model1 and Model2. Model4, while having a slightly lower P than Model1, showed the best performance in terms of R, mAP@0.5, and mAP@0.5:0.95. Model1 achieved the best P due to its use of the SIOU (Scale Sensitive Intersection over Union) loss function [53] instead of the original CIOU loss function. The SIOU loss function introduces vector angles between the desired regressions, redefines the distance loss, effectively reduces the degree of regression freedom, accelerates network convergence, and improves the regression accuracy. In the future, we will introduce SIOU into our model for further improvement.

However, the selection of appropriate improvement modules is also crucial for improving the original deep learning models. Therefore, in this study, we conducted experiments to assess the suitability of the selected ECA and SPPFCSPC modules.

The results of the ablation experiments show that the models with the ECA module and SPPFCSPC module which were introduced both improve on the performance of the original YOLOv5s-seg model. The FFDSM significantly improves performance over the baseline YOLOv5s-seg model in object detection and segmentation tasks. It performs excel-

lently across different evaluation metrics, with the most notable improvement observed in mAP@0.5:0.95, which is increased by 4.2%. This indicates that the combination of the introduced modules effectively enhances the model's performance.

**Table 11.** Selected advanced algorithm comparison.

| Number | Model | P | R | mAP@0.5 | mAP@0.5:0.95 |
|--------|-------|---|---|---------|--------------|
| 1 | [43] (SIOU+CBAM+BiFPN) | **0.961** | 0.846 | 0.893 | 0.685 |
| 2 | [44] (CA+RFB+BiFPN) | 0.947 | 0.856 | 0.894 | 0.692 |
| 3 | [10] (CBAM+SPPFCSPC+BiFPN) | 0.951 | 0.866 | 0.899 | 0.701 |
| 4 | FFDSM | 0.959 | **0.87** | **0.907** | **0.711** |

Note: CBAM, CA represents Convolutional Block Attention Module, Coordinate Attention, respectively; BiFPN [54]: Bidirectional Feature Pyramid Network; RFB [55]: Receptive Field Block Spatial Pyramid Pooling. Bold numbers represent perform best in each metrics.

The controlled experiments of the different attention modules and pyramid modules show that ECA+SPPFCSPC outperforms the other combinations of attention mechanisms and pyramid pooling modules, achieving the highest scores across multiple evaluation metrics. It achieves a P of 0.959, R of 0.870, mAP@0.5 of 0.907, and mAP@0.5:0.95 of 0.711. This indicates that ECA+SPPFCSPC has significant advantages in object detection and segmentation tasks, as it can better extract features, utilize the contextual information in images, and demonstrate superior contextual modeling capabilities.

The controlled experiments on different fire sizes show that the FFDSM consistently outperforms the original YOLOv5s-seg model for all three fire sizes, with a P reaching above 0.95. This demonstrates that the FFDSM is well-suited for detecting fires of different sizes. Additionally, the FFDSM performs best in detecting small fires, indicating its superior applicability in early forest fire detection, allowing for the more accurate detection and localization of small fires.

According to the ablation and controlled experiments, we noticed that the FFDSM has the highest improvement in mAP@0.5:0.95 and also shows improvements for different fire sizes. This can be attributed to the SPPFCSPC module, which performs four different MaxPool operations representing different scales of receptive fields. This enables the model to better distinguish between large and small objects, resulting in improved generalization. In addition, the ECA module uses a $1 \times 1$ convolutional layer after the global average pooling layer and removes the fully connected layer. This module avoids dimensionality reduction and efficiently captures cross-channel interactions. In addition, the ECA module completes cross-channel information interactions through one-dimensional convolution. The size of the convolution kernel adapts to changes through a function so that layers with a larger number of channels can perform more cross-channel interactions.

As shown in Figure 8, both YOLOv5s-seg and the FFDSM accurately detect large fires. However, the FFDSM outperforms YOLOv5s-seg in terms of accuracy, especially in the case of small fires. This high-precision detection, specifically for small fires, is crucial for fire detection and response. Small fires often serve as the first sign of a fire in its early stages. Identifying and locating these small fires promptly can facilitate the rapid implementation of effective firefighting measures, preventing further escalation and spread of the fire. Therefore, the high-precision detection capability of the FFDSM model holds significance for enhancing fire detection capabilities.

While using infrared imagery for forest fire detection can effectively address the limitations of optical imagery, there is currently a scarcity of forest fire datasets based on infrared imagery, and the available datasets often represent limited forest fire scenarios [56,57]. Therefore, we will conduct UAV experiments in the future to obtain forest fire optical and infrared images in different scenarios and further improve the detection accuracy of forest fires in different scenarios through the fusion of visible light and infrared images. In particular, for small fires, we will further optimize the FFDSM model to achieve the small target detection of forest fires.

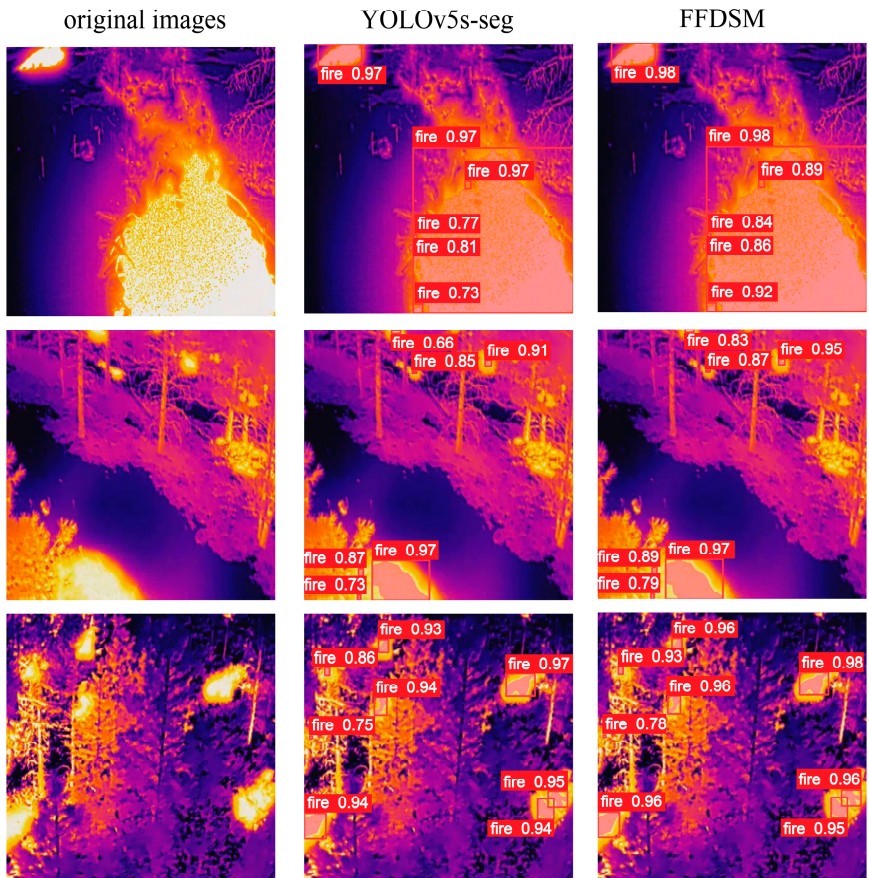

**Figure 8.** Comparison results of YOLOv5s-seg and FFDSM. (The first column is original image; the second column is the detection result of YOLOv5s-seg; the third column is the detection result of the FFDSM).

## 5. Conclusions

The study presents the FFDSM, a forest fire detection and segmentation method based UAV infrared images, which addresses the problems of forest fire occlusion and the poor adaptability traditional forest fire detection methods. The FFDSM model incorporates the ECA module and SPPFCSPC module.

In this study, we conducted various experiments to evaluate the impacts of the improvement modules in the FFDSM and its performance under different fires. These experiments included ablation experiments and three controlled experiments with different attention mechanisms, different spatial pyramid pooling modules, and different fire sizes. The conclusions are as follows:

(1) The ablation experiment results demonstrated that the inclusion of the ECA module and SPPFCSPC module improved the accuracy of the original YOLOv5s-seg model. Among the models, the FFDSM achieved the greatest improvements, with increases of 2.1% in precision (P), 2.7% in recall (R), 2.3% in mAP@0.5, and 4.2% in mAP@0.5:0.95.

(2) The results of the controlled experiments on different attention mechanisms and spatial pyramid pooling modules showed that the ECA+SPPFCSPC model performed the best in all the evaluation metrics, with a P, R, mAP@0.5, and mAP@0.5:0.95 reaching 0.959, 0.870, 0.907, and 0.711, respectively.

(3) The results of the controlled experiments on different fire sizes showed that the FFDSM outperformed YOLOv5s-seg for all three fire sizes, indicating that it has better applicability. In addition, the FFDSM performed best in the case of small fires; thus, it can be used for early forest fire detection.

(4) In conclusion, the proposed forest fire detection and segmentation method based on UAV infrared imagery and the FFDSM exhibits a high accuracy and fast detection speed. It enables the real-time detection of occluded forest fires and provides early warning of forest fires, providing reliable support for forest fire prevention and scientific decision making.

**Author Contributions:** Conceptualization, K.N. and X.Z.; formal analysis, K.N.; data curation, K.N.; writing—original draft preparation, K.N.; writing—review and editing, X.Z., C.W., J.X., X.Y. and C.Y. All authors have read and agreed to the published version of the manuscript.

**Funding:** This research was supported by Guangdong Forestry Science and Technology Innovation Project (2021KJCX020).

**Data Availability Statement:** The data presented in this study are available on request from the first or corresponding author.

**Acknowledgments:** The authors are very grateful to the Northern Arizona University for providing the FLAME dataset: Aerial Imagery Pile Burn Detection Using Drones (UAVs).

**Conflicts of Interest:** The authors declare no conflict of interest.

## Abbreviations

| Abbreviations | Full name | Description |
| --- | --- | --- |
| BiFPN | Bidirectional Feature Pyramid Network | neural network structure |
| BN | Batch Normalization | components of CBS |
| CA | Coordinate Attention | attention mechanism |
| CBAM | Convolutional Block Attention Module | attention mechanism |
| CBS | Cov, BN, SiLU | neural network module |
| CIOU | Complete Intersection over Union | loss function |
| CMOS | Complementary Metal-Oxide Semiconductor | camera sensor |
| CNN | Convolutional Neural Networks | neural network |
| Cov/Conv | Convolutional | convolutional module |
| CSP | Cross Stage Partial | neural network structure |
| ECA | Efficient Channel Attention | attention mechanism |
| EO-1 | Earth Observing-1 | earth observation satellites |
| FFDSM | Forest Fire Detection and Segmentation Model | forest fire detection model |
| FLAME | Fire Luminosity Airborne-based Machine learning Evaluation | dataset |
| FOV | Field of View | camera parameters |
| FPN | Feature Pyramid Network | neural network structure |
| GIOU | Generalized Intersection over Union | loss function |
| IOU | Intersection over Union | evaluation metric |
| IR | Infrared | — |
| L | large | — |
| M | medium | — |
| MaxPool | Maximum Pooling | neural network module |
| MODIS | Moderate-resolution Imaging Spectroradiometer | satellite sensors |
| P | precision | evaluation metric |
| PAN | Pyramid Attention Network | neural network structure |
| R | recall | evaluation metric |
| R-CNN | Region-based Convolutional Neural Networks | object detection model |
| RFB | Receptive Field Block Spatial Pyramid Pooling | pyramid pooling module |
| RGB | Red, Green, Blue | color model |
| S | small | — |
| SE | Squeeze and Excitation | attention mechanism |
| SGD | Stochastic Gradient Descent | model training parameters |

| SIOU | Scale Sensitive Intersection over Union | loss function |
|---|---|---|
| SPP | spatial pyramid pooling | cmpyramid pooling module |
| SPPCSPC | Spatial Pyramid Pooling Cross-Stage Partial Channel | pyramid pooling module |
| SPPF | Spatial Pyramid Pooling Fast | pyramid pooling module |
| SPPFCSPC | Spatial Pyramid Pooling Fast Cross-Stage Partial Channel | pyramid pooling module |
| SSD | Single Shot Multi-box Detector | object detection model |
| UAV | unmanned aerial vehicle | — |
| VIIRS | Visible infrared Imaging Radiometer | satellite sensors |
| X | extra-large | — |
| Ycbcr | Luminance, Chrominance-Blue, Chrominance-Red | color model |
| YOLO | You Only Look Once | object detection model |

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
