# Peer review of "An Improved YOLOv5s-Seg Detection and Segmentation Model for the Accurate Identification of Forest Fires Based on UAV Infrared Image"

_remotesensing, doi:10.3390/rs15194694_

Round 1

Reviewer 1 Report

Title: An improved YOLOv5s-seg detection and segmentation model for the accurate identification of forest fires based on UAV infrared image

Overview:

The study compares approaches to remote sensing based fire detection, but is full of jargon and acronyms that might not be familiar to many readers. More information should be included to explain the datasets used (even though there is a link to the source), explain what was actually done, and define ALL acronyms used in the paper. None of the table or figure captions contain sufficient information. 

Other comments:

Authors should define all acronyms the first time used in the paper. E.g., I don’t see anything defining YOLO or CNN. Many others. 

Figures 2 and 3 contain many acronyms- including those that were never defined in the text. Caption should explain. 

Figure 4 needs a longer caption to explain what is shown in the graphic.

Section 2.4 Sources for the metrics used in model evaluation? 

Tables need more information in the captions. All acronyms and abbreviations should be defined. 

Author Response

Thank you for your comments. We have revised the manuscript. Please see the attachment.

Reviewer 2 Report

This study proposes a Forest Fire Detection and Segmentation Model (FFDSM) for forest fire detection based on infrared images from unmanned aerial vehicles (UAV) to solve the problems of forest fire occlusion and the failure in the adaptability of traditional methods of detection. detection of forest fires.

- The article presents an interesting investigation with processing and positive results, but in my opinion the article needs several technical-scientific improvements to be published in this journal.

- A literature review needs to be improved. There are no bibliographic references of studies similar to satellite images of high spatial resolution. It is also necessary to include some more bibliographic instructions on the methodological procedures used and some results obtained. We consider the number of bibliographical references of 36 for this article to be small.

- There is no detailing of the specifications of the sensor system (UAV) and the aerial survey used. There is a lack of technical information regarding the flight plan and specifications of the sensor (UAV) used.

What is the spatial resolution of the system (UAV) used?

What is the spectral resolution of the system (UAV) used?

- A better description of the materials and methods used in the study is needed in the spatial and spectral part. The infrared spectral range covers parts of the optical and thermal spectrum. I believe that this study was carried out in the optical infrared spectral range. Specify the spectral bands used with their respective measurement units.

- The size of the pixel under study is much smaller than that of the Modis sensor. So I don't believe the comment with the Modis sensor is entirely pertinent.

- I think the article would be more consistent if it were included in an analysis of the influence of spectral resolution in the detection of burned areas. Then we would have the spatial and spectral part.

- The article has technical-scientific potential. This article, after being corrected and improved within the scientific context, will have a great chance of being approved for publication in this journal.

Author Response

(The authors gave the same response as above.)

Reviewer 3 Report

The abstract is written well. The literature review part should be improved. In a few instances, the authors stated several technicalities without reference that need to be modified as well.

Although it is a technical development-oriented work, I suggest giving a little more emphasis on scientific aspects in terms of its applicability to real scenarios. For example, the authors could segregate the entire images into three parts - ground fire, trunk fire and crown fire and then exhibit the model performance for each of these.  

The authors must mention how large and small fires are classified from an image. Have they calculated the area from the images, or was it based on pixel counts covering the fire portion? In the latter case, is the pixel size or image resolution the same for each image?

Forest fires can be at ground level (limited to litter) or reached up to a crown level. Often, it is found that ground fires are not able to be detected. Does the current proposed model encounter any such scenario? 

Usually, thermal infrared images have a coarser resolution than optical images. In this case, how efficiently small-scale ground fires can be detected by using the proposed model?

Forest fire attributes are different in varying topography. Are the sample images taken from all different topography?

Line 148-149, add the advantage of SiLu over Leaky ReLu.

Since you are mentioning [email protected]:0.95, it is not required to further mention [email protected].

There are a few grammatical and spelling mistakes which need to be corrected.

Author Response

(The authors gave the same response as above.)

Round 2

Reviewer 1 Report

Figures 3, 4, 5, 6 are still not clear to me- even with the added description in the caption. 

Sections 2.2.1, 2.2.2, and 2.2.3 still seem to contain a lot of jargon that is difficult to follow. 

Author Response

(The authors gave the same response as above.)

Reviewer 2 Report

The suggested and implemented changes made the article more consistent. I consider the article approved for publication.

Author Response

Thanks for your constructive suggestions. We appreciate your support of our work, which makes us full of confidence for futher research.
